# Investigating effect modification between childhood maltreatment and genetic risk for cardiovascular disease in the UK Biobank

Helena Urquijo[1,2]*, Ana Gonçalves Soares[1,2], Abigail Fraser[1,2], Laura D. Howe[1,2], Alice R. Carter[1,2]

1 Medical Research Council Integrative Epidemiology Unit, University of Bristol, Bristol, United Kingdom,
2 Population Health Sciences, Bristol Medical School, University of Bristol, Bristol, United Kingdom

* helena.urquijo@bristol.ac.uk

**Data Availability Statement:** All relevant data are within the paper and its Supporting Information files.

## Abstract

Cardiovascular disease (CVD) is influenced by genetic and environmental factors. Childhood maltreatment is associated with CVD and may modify genetic susceptibility to cardiovascular risk factors. We used genetic and phenotypic data from 100,833 White British UK Biobank participants (57% female; mean age = 55.9 years). We regressed nine cardiovascular risk factors/diseases (alcohol consumption, body mass index [BMI], low-density lipoprotein cholesterol, lifetime smoking behaviour, systolic blood pressure, atrial fibrillation, coronary heart disease, type 2 diabetes, and stroke) on their respective polygenic scores (PGS) and self-reported exposure to childhood maltreatment. Effect modification was tested on the additive and multiplicative scales by including a product term (PGS*maltreatment) in regression models. On the additive scale, childhood maltreatment accentuated the effect of genetic susceptibility to higher BMI ($P_{effect\ modification}$: 0.003). Individuals not exposed to childhood maltreatment had an increase in BMI of 0.12 SD (95% CI: 0.11, 0.13) per SD increase in BMI PGS, compared to 0.17 SD (95% CI: 0.14, 0.19) in those exposed to all types of childhood maltreatment. On the multiplicative scale, similar results were obtained for BMI though these did not withstand to Bonferroni correction. There was little evidence of effect modification by childhood maltreatment in relation to other outcomes, or of sex-specific effect modification. Our study suggests the effects of genetic susceptibility to a higher BMI may be moderately accentuated in individuals exposed to childhood maltreatment. However, gene*environment interactions are likely not a major contributor to the excess CVD burden experienced by childhood maltreatment victims.

## Introduction

Cardiovascular disease (CVD) is the leading cause of death worldwide [1]. Many CVD subtypes are accepted to have complex aetiologies that are affected by both genetic and environmental factors [2, 3]. This suggests that interactions between genetic and environmental factors may play a role in disease development and burden.

**Funding:** HU is funded by a PhD studentship from the British Heart Foundation (FS/17/60/33474) (https://www.bhf.org.uk/what-we-do/our-research). HU, ALGS, AF, LDH and ARC all work in a unit that receives core funding from the UK Medical Research Council and University of Bristol (MC_UU_00011/1, MC_UU_00011/4 and MC_UU_00011/6) (https://www.ukri.org/councils/mrc/). ARC is additionally supported by the University of Bristol British Heart Foundation Accelerator Award (AA/18/7/34219). ALGS is supported by the study of Dynamic longitudinal exposome trajectories in cardiovascular and metabolic non-communicable diseases (H2020-SC1-2019-Single-Stage-RTD, project ID 874739) (https://cordis.europa.eu/project/id/874739). LDH is funded by a Career Development Award from the UK Medical Research Council (MR/M020894/1). The funders had no role in study design, data collection and analysis, decision to publish, or preparation of the manuscript.

**Competing interests:** The authors have declared that no competing interests exist.

Childhood maltreatment increases the risk of a number of CVD subtypes and related diseases, including coronary heart disease (CHD), stroke and type 2 diabetes mellitus (T2DM) [4, 5]. Evidence also suggests there is a dose-response relationship between childhood maltreatment and CVD, with disease risk increasing according to the number of types of maltreatment experienced by an individual [6–13]. While this growing body of evidence indicates childhood maltreatment is associated with higher CVD risk, it is unknown whether this risk is disproportionate in genetically susceptible individuals.

Single nucleotide polymorphisms (SNPs) detected in genome-wide association studies (GWAS) have been used to construct polygenic scores (PGSs) for a trait or disease. These estimate the genetic risk carried by an individual by adding the effects of all relevant variants in the genome. The polygenic nature of many diseases, including CVD, suggests these scores may be more informative when investigating interactions or effect modification by environmental factors, in comparison to using a single associated gene [14]. PGSs have previously been employed to assess gene*environment interactions or effect modifications for cardiovascular traits, including for educational achievement, neighbourhood and obesogenic environments [15–17]. Identifying these effects may help to understand the relationship between early life environmental exposures and CVD risk in adult life.

This study aims to investigate whether childhood maltreatment modifies genetic susceptibility to a range of cardiovascular risk factors and CVD subtypes, using data from the UK Biobank cohort. Where two risk factors are associated with the outcome, evidence of effect modification is expected on either the additive or the multiplicative scale, or on both scales [18, 19]. Therefore, in this analysis we investigate both scales. Of greatest relevance to public health is the direction and magnitude of any interaction. We hypothesise that childhood maltreatment accentuates the effect of polygenic predisposition to CVD subtypes and risk factors. We also investigate whether any potential effect modification differs according to sex, as some studies have detected differences in the association between childhood maltreatment and cardiovascular risk [13, 20–23].

## Methods

### UK Biobank

UK Biobank recruited 503,317 adults aged 37–73 years at baseline, between 2006 and 2010 [24]. At baseline, data was collected on health and lifestyle information, physical measurements and biological samples. Some follow-up questionnaires and clinics have taken place, including an online mental health questionnaire in 2016, which asked about childhood maltreatment. This was sent to participants who provided an email address (N = 339,092) and 157,366 (46.4%) responses were obtained. This analysis includes 100,833 participants of White British ancestry who completed questions on childhood maltreatment and had genetic information. Some individuals were missing CVD risk factor information and thus sample sizes for analyses ranged from n = 93,001 to n = 100,833 (Fig 1).

### Childhood maltreatment

The types of maltreatment assessed comprised of emotional neglect, physical neglect, emotional abuse, physical abuse and sexual abuse [25, 26]. A summary score of childhood maltreatment was calculated as per the Childhood Trauma Screener [25]. The score reflects the cumulative number of maltreatment types experienced by an individual, spanning from 0 to 5, according to cut-offs specific to each type (S1 Table in S1 File).

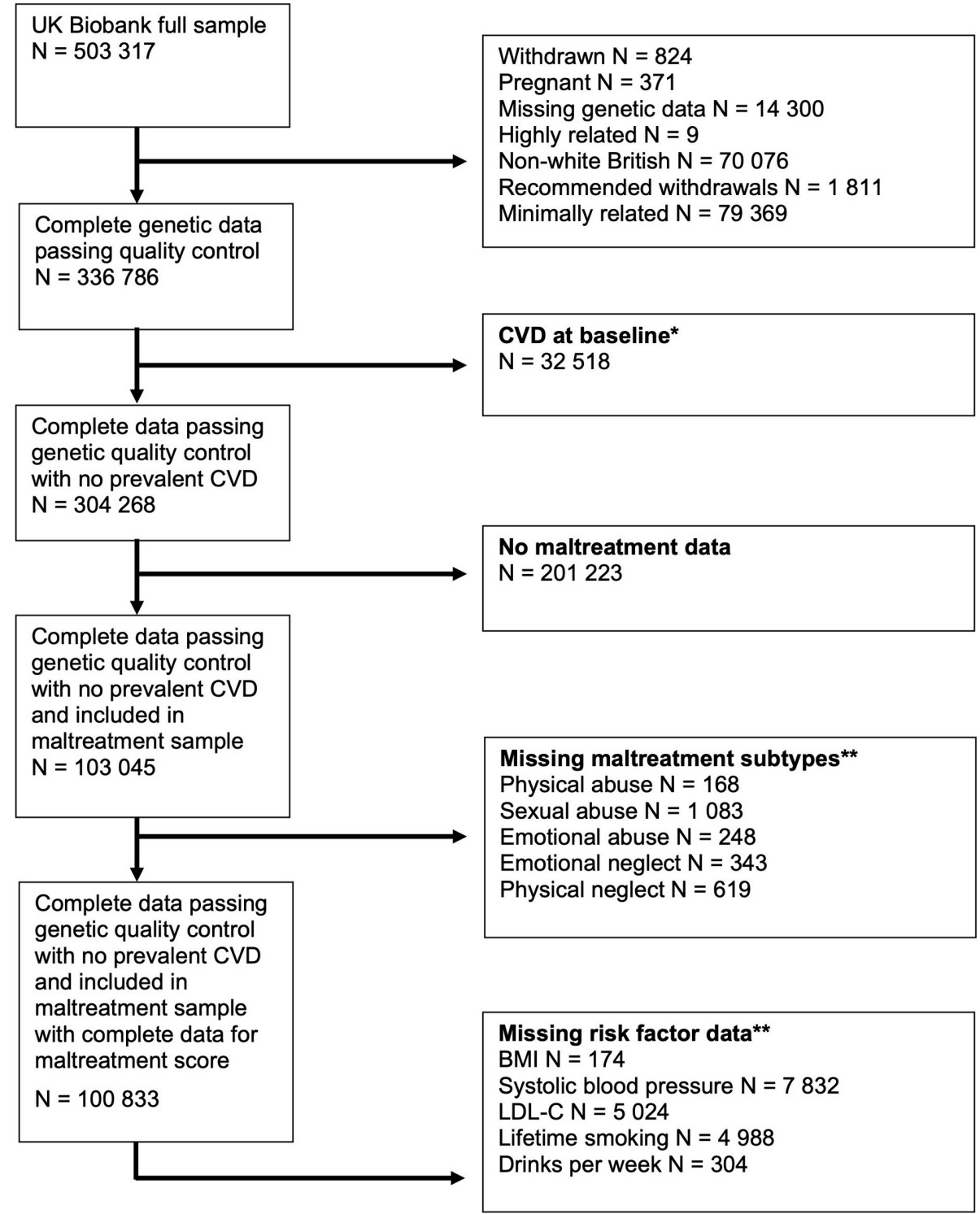

**Fig 1. Study exclusion criteria.** *CVD = stroke, atrial fibrillation, coronary heart disease or risk factor type 2 diabetes. **Participants can be missing more than one variable, so the total excluded is less than the sum of the missing data of all traits.

## Cardiovascular risk factors and diseases

This study included 3 CVD subtypes and 6 risk factors for CVD. Risk factors were considered if there is evidence for a causal association with CVD from randomised control trials (RCT) or Mendelian randomisation studies (see S2 Table in S1 File )—a statistical approach using randomly allocated genetic variants as instrumental variables [27]—and have GWAS information with summary statistics available.

Quantitative traits (body mass index (BMI), alcohol consumption measured as drinks per week, low-density lipoprotein cholesterol (LDL-C), lifetime smoking behaviour and systolic blood pressure (SBP) were determined at baseline through physical or biological measurements, or self-reported for alcohol and smoking behaviours.

Incidence of CVD subtypes (atrial fibrillation (AF), CHD and stroke) and of risk factor T2DM were determined through linked mortality data, hospital episode statistics and Scottish morbidity and mortality records (SMR), defined using ICD-9 and ICD-10 codes (S3 Table in S1 File).

The period of follow-up occurred between the date of baseline assessment (between 2006 and 2010) and the most recent available linked hospital inpatient data, from May 2017.

## Polygenic scores

Summary statistics were obtained from the most recent GWAS (Table 1). Where possible, if the most recent GWAS included UKB participants, these were excluded to avoid estimate inflation through sample overlap (all GWAS are independent of UKB except for AF). PGSs were calculated by multiplying the number of effect alleles by the relative effect estimate of the SNP on the phenotype and summing together the alleles.

PGSs were generated at P value thresholds of $<5 \times 10^{-8}$ (genome-wide significance), $<0.05$ and $<0.5$. The greater the P value threshold for SNPs, the greater the variance explained by the score (S4 Table in S1 File), but the greater the chance of capturing pleiotropic effects. Resulting PGSs were standardised, reflecting a one standard deviation increase in PGS.

Full details of the derivation of PGSs, GWAS studies used and generation of phenotypic measures can be found in the S1 File.

## Exclusion criteria

Quality control of UK Biobank genetic data was performed in accordance with the MRC-IEU Quality Control pipeline, as previously described [37]. In brief, participants with sex mismatch (comparing genetic sex with reported sex) and those with sex chromosome aneuploidy were excluded from the study. Participants of White British ancestry were those who self-reported as "White British" and supported by genetic principal component (PC) analysis. Related participants were excluded according to kinship coefficients provided by the UK Biobank until no related pairs remained [37].

Individuals missing age, sex, genetic data or who did not have complete information for childhood maltreatment were excluded from the sample (Fig 1). Individuals missing risk factor or disease data were excluded from relevant analyses. To prevent the detection of effects caused by reverse causality, participants were excluded if they experienced any diagnosis of AF, T2DM, CHD or stroke prior to baseline. Prevalent cases were ascertained by comparing the date of diagnosis in the linked hospital inpatient data to the date of baseline visit.

**Table 1. Summary characteristics for each GWAS used to derive external weights in polygenic scores.**

| Phenotype | Author/consortium | Population | Sample Size (cases) | Unit |
|---|---|---|---|---|
| Alcohol consumption | GWAS and Sequencing Consortium of Alcohol and Nicotine use [28] | European (Summary statistics excluding UK Biobank) | 630 154 | Drinks per week |
| Body mass index | Genetic Investigation of Anthropometric Traits [29] | European ancestry | 339 224 | SD (kg/m^2) |
| Low-density lipoprotein cholesterol | Global Lipids Genetics consortium [30] | European ancestry | 188 578 | SD (circulating lipids) |
| Smoking | Wootton *et al* [31] | White British (split sample GWAS of UK Biobank, see Supplementary Methods in S1 File) | 318 147 | SD (Lifetime smoking index) |
| Systolic blood pressure | Carter *et al* [32] | White British (split sample GWAS of UK Biobank, see Supplementary Methods in S1 File) | 318 147 | SD (mm/Hg) |
| Atrial fibrillation | Christophersen *et al* [33] | European ancestry | 154 432 (22 346) | Log odds ratio |
| Coronary heart disease | CARDIoGRAMplusC4D [34] | Predominantly European (77%) | 184 305 (60 801) | Log odds ratio |
| Type 2 diabetes | DIAbetes Genetics Replication And Meta-analysis [35] | European ancestry | 159 208 (26 276) | Log odds ratio |
| Stroke | MEGASTROKE [36] | Predominantly European (85%) | 521 612 (67 162) | Log odds ratio |

## Statistical analysis

**Data and code availability.** Statistical analyses were carried out using Stata16-MP. Analysis code is available at github.com/helenaurquijo/gxe_cvd_maltreatment. Study data used will be archived with UK Biobank and was carried out under the approved UK Biobank project 19278. No individual patient consent was required.

**Association of childhood maltreatment with outcomes.** Multivariable linear and logistic regression models, adjusting for sex and age, were used to estimate the association between the childhood maltreatment score and each of the nine risk factors/diseases.

**Effect modification of childhood maltreatment on genetic susceptibility.** For continuous traits, the association of childhood maltreatment score and each PGS with the respective trait was estimated on the additive scale via multivariable linear regression. On the multiplicative scale, the association was estimated via multivariable linear regression for the natural log of the trait.

For binary traits, the association of childhood maltreatment score and each PGS with the respective disease trait was estimated on the additive scale via multivariable linear regression. On the multiplicative scale, the association was estimated via multivariable logistic regression.

All regressions were adjusted for sex, age and 40 genetic PCs. Continuous phenotypic measures of traits were standardised to reflect a change in SD of trait per 1 SD increase in PGS. Coefficients for binary traits reflect a change in risk per 1 SD increase in PGS.

To test for effect modification of genetic predisposition by exposure to childhood maltreatment, the same regression models were run including a product of the PGS and maltreatment score as an interaction term. Effect modification was evaluated by the size and precision of the coefficient for effect modification, and the P value for the effect modification coefficient considering Bonferroni correction.

To address the risk of type 1 error due to multiple comparisons, a Bonferroni correction was applied by dividing the standard 0.05 P value threshold by the number of outcomes we considered in each analysis (nine), yielding a threshold of P = 0.006. The use of this correction approach was conservative as the considered traits are not independent from each other.

**Secondary analyses.** Sex-stratified analyses were carried out, and a 3-way interaction term (i.e. PGS\*maltreatment\*sex) on the respective risk factor or CVD subtype was included.

Main analyses were repeated using PGSs generated with less stringent P value thresholds at 0.05 and 0.5, as described previously.

Finally, sensitivity analyses were performed by replicating the main analyses with adjustments for covariate interactions (e.g. sex\*PGS, age\*maltreatment. . .) to avoid specification error [14]. Separately, maternal smoking and number of siblings (proxies of childhood socio-economic position), which may confound associations between childhood maltreatment and CVD, were included in the adjustment.

### Ethics statement

UKB has approval from the North West Multi-centre Research Ethics Committee (MREC) as a Research Tissue Bank (RTB) approval. All analyses were performed under approved UKB project 19278. As part of the UKB recruitment process, all necessary patient/participant consent was obtained and the appropriate institutional forms were archived. Data were analysed anonymously.

## Results

### UK Biobank sample

Eligible participants had a mean age at baseline of 56 years (SD: 8.0 years) and 57% were female. Overall maltreatment scores did not differ largely between sexes, yet more female participants reported sexual and emotional abuse and less physical abuse compared to male participants (Table 2).

Most participants (55%) reported no exposure to any type of childhood maltreatment. The most frequently reported type of maltreatment was emotional neglect (21%) and sexual abuse was the least reported (8%) (S5 Table in S1 File). Our study sample did not differ from the rest of the UK Biobank in terms of exposure to maltreatment, cardiovascular risk and cases of CVD.

With the genome-wide significance threshold ($P<5x10^{-8}$), variance explained by the PGSs ranged between 0.06% (smoking) and 15% (systolic blood pressure) (S4 Table in S1 File).

### Association of childhood maltreatment with cardiovascular risk factors and disease

Childhood maltreatment score was associated with higher BMI (β: 0.07 SD; 95% CI: 0.06, 0.07), where 1 SD corresponds to 4.38 kg/m$^2$, and lifetime smoking, as well as with T2DM and CHD incidence (S5 Table in S1 File). However, childhood maltreatment was negatively associated with systolic blood pressure (β: -0.011 SD; 95% CI: -0.017, -0.005). Childhood maltreatment was positively associated with all other outcomes, but the confidence intervals spanned the null.

### Effect modification by childhood maltreatment

On the additive scale, there was little evidence that childhood maltreatment modified the effect of the PGSs for all risk factors/diseases except for BMI (S6 Table in S1 File and Fig 2). The effect of genetic susceptibility to higher BMI was accentuated by childhood maltreatment score by 0.009 SD (95% CI: 0.003, 0.014; P$_{effect modification}$: 0.003). This indicates, in those unexposed to childhood maltreatment, each SD increase in the BMI PGS was associated with 0.12 SD higher BMI (95% CI: 0.11, 0.13), whilst in those with a maltreatment score of 5, each SD

**Table 2. Descriptive statistics of study sample, stratified by sex.**

| Variable | Female N = 57 641 | | | Male N = 43 192 | | |
|---|---|---|---|---|---|---|
| **Continuous** | N | Mean (SD) | | N | Mean (SD) | |
| Age | 57 641 | 55.51 (7.58) | | 43 192 | 56.37 (7.76) | |
| BMI | 57 540 | 26.19 (4.71) | | 43 119 | 27.07 (3.83) | |
| Drinks per week | 57 541 | 6.36 (6.92) | | 42 988 | 11.07 (9.93) | |
| LDL cholesterol | 54 698 | 3.65 (0.83) | | 41 111 | 3.61 (0.80) | |
| Lifetime smoking | 54 768 | 0.22 (0.52) | | 41 077 | 0.28 (0.59) | |
| Systolic blood pressure | 53 143 | 133.46 (18.47) | | 39 858 | 140.26 (16.79) | |
| Number of siblings | 56 953 | 1.79 (1.46) | | 42 676 | 1.78 (1.49) | |
| **Categorical** | N | Frequency (%) | | N | Frequency (%) | |
| Maternal smoking | 51 102 | No | 36 620 (72) | 37 805 | No | 26 733 (71) |
| Atrial fibrillation (incident) | 57 641 | Case | 771 (1) | 43 192 | Case | 1 327 (3) |
| Type 2 Diabetes (incident) | 57 641 | Case | 566 (1) | 43 192 | Case | 818 (2) |
| Coronary Heart Disease (incident) | 57 641 | Case | 1 029 (2) | 43 192 | Case | 2 003 (5) |
| Stroke (incident) | 57 641 | Case | 439 (1) | 43 192 | Case | 556 (1) |
| Physical abuse | 100 833 | Yes | 9 070 (16) | 156982 | Yes | 8 460 (19) |
| Emotional abuse | | | 9 432 (16) | 156886 | | 5 103 (12) |
| Sexual abuse | | | 5 827 (10) | 155488 | | 2 364 (6) |
| Physical neglect | | | 9 307 (16) | 156266 | | 5 979 (14) |
| Emotional neglect | | | 12 394 (22) | 156712 | | 8 795 (20) |
| Childhood maltreatment score | 57 641 | 0 | 31 840 (55) | 43 192 | 0 | 24 070 (56) |
| | | 1 | 13 980 (24) | | 1 | 11 590 (27) |
| | | 2 | 6 224 (11) | | 2 | 4 622 (11) |
| | | 3 | 3 360 (6) | | 3 | 1 939 (4) |
| | | 4 | 1 663 (3) | | 4 | 805 (2) |
| | | 5 | 574 (1) | | 5 | 166 (0) |

For maltreatment types, exposure is determined according to cut-off criteria described in S1 Table in S1 File.

increase in the PGS was associated with 0.17 SD higher BMI (95% CI: 0.14, 0.19). All other interaction effect estimates were positive, with the exception of SBP, and extremely small.

On the multiplicative scale, similar yet less precise results were obtained (S7 Table in S1 File and Fig 3). The effect of genetic susceptibility to higher log BMI was accentuated by childhood maltreatment, through an increase of 0.007 SD (95% CI 0.001, 0.013) per unit increase in maltreatment score. Nonetheless, the interaction estimate did not withstand Bonferroni correction ($P_{effect\ modification}$: 0.015). The coefficient for effect modification was in the positive direction for alcohol consumption, LDL-C, smoking, AF and stroke, and in the negative direction for SBP, CHD and T2DM. However, for all outcomes the estimates were close to the null and confidence intervals were narrow.

## Secondary analyses

Sex was not found to modify the effects observed in the main analyses on either the additive or the multiplicative scale (S8 and S9 Tables in S1 File).

Analyses repeated with PGSs generated at P value thresholds of P≤0.05 and P≤0.5 found childhood maltreatment score to accentuate the effect of genetic susceptibility to higher BMI and greater lifetime smoking, on both the additive and multiplicative scale (S10 and S11 Tables in S1 File).

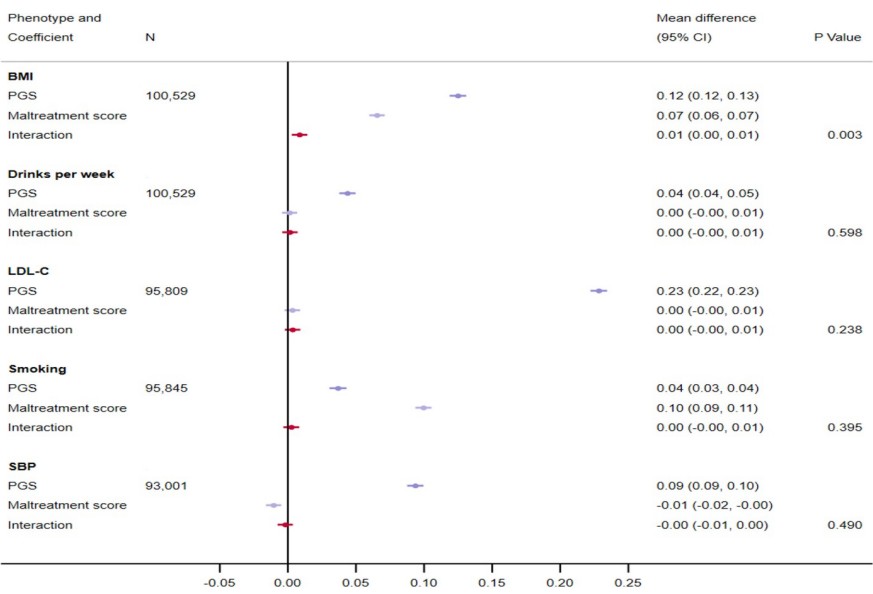

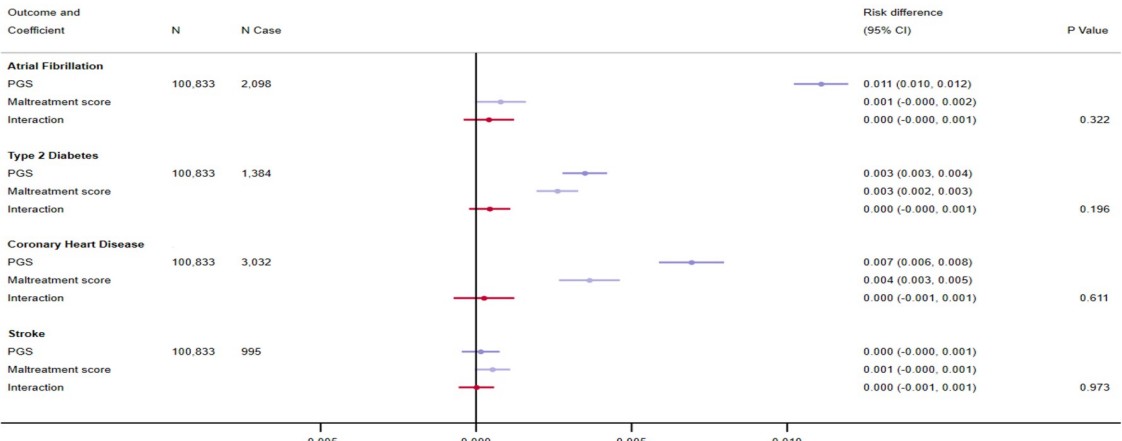

**Fig 2. Effect modification by exposure to childhood maltreatment on genetic predisposition to cardiovascular risk factors and disease on the additive scale at P<5x10⁻⁸.** Analyses were adjusted for age, sex and 40 genetic principal components. Traits are grouped according to whether they are binary or continuous. BMI = body mass index; LDL-C = low-density lipoprotein cholesterol; SBP = systolic blood pressure; PGS = polygenic score.

Analyses adjusting for covariate (age, sex and genetic PCs) interactions with childhood maltreatment score and PGSs yielded similar results, though the interaction coefficients for BMI became less precise, decreasing the strength of evidence for effect modification on the multiplicative scale (S12 Table in S1 File).

Analyses adjusting for maternal smoking and number of siblings yielded similar results to main analyses (S13 Table in S1 File).

## Discussion

We found evidence that childhood maltreatment accentuates the effect of polygenic susceptibility to BMI on observed BMI but no other cardiovascular risk factors or outcomes

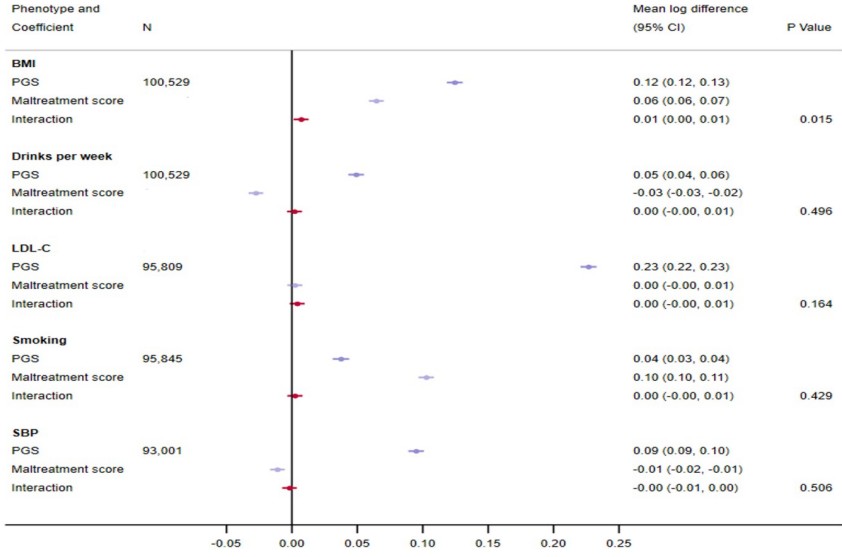

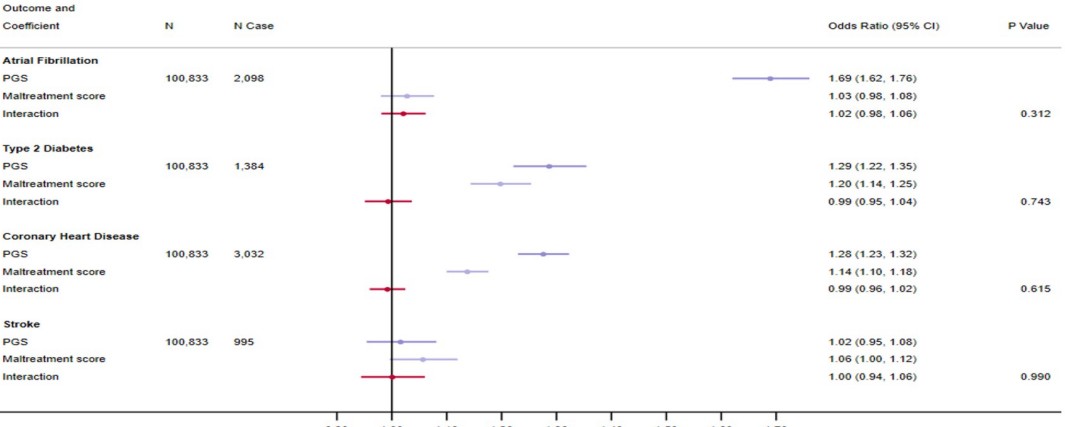

**Fig 3. Effect modification by exposure to childhood maltreatment on genetic predisposition to cardiovascular risk factors and disease on the multiplicative scale at P<5x10⁻⁸.** Analyses were adjusted for age, sex and 40 genetic principal components. Traits are grouped according to whether they are binary or continuous. BMI = body mass index; LDL-C = low-density lipoprotein cholesterol; SBP = systolic blood pressure; PGS = polygenic score.

considered. Effects were consistent across the additive and multiplicative scales, though evidence of effect modification for BMI was weaker on the multiplicative scale. Additionally, we found no evidence of sex differences in these effects.

## Results in context

Few studies have investigated whether childhood maltreatment interacts with or modifies the effect of genetic susceptibility to a range of cardiovascular risk factors/diseases. In a candidate gene approach, Gooding and colleagues found no association between childhood maltreatment and hypertension in young adulthood, nor the presence of effect modification by the SLC64A genotype [38]. In our study of adults, we found a negative association between childhood maltreatment and SBP. In a study with 161 individuals with major depression, Opel and

colleagues also found childhood maltreatment modified polygenic susceptibility to BMI but in the opposite direction, with exposure to maltreatment reducing the effect of polygenic susceptibility on BMI [39]. These opposing findings could be due to different definitions of maltreatment (binary versus continuous score), or different study populations (depressive patients versus population cohort). Contrastingly, a recent study found midlife BMI to be the major mediator in the association between early life abuse and midlife metabolomic profile [40]. These results, despite being specific to women, coupled with our findings, are in support of a potential role of childhood maltreatment in adverse metabolic outcomes in later life.

We expected to detect effect modification for all outcomes given (a) the association between PGSs and respective phenotypes and (b) the positive associations between childhood maltreatment and CVD subtypes present in our study sample (see S4 Table in S1 File and interpreting interactions section below). It is difficult to determine whether the lack of effect modification for most outcomes is truly null, underpowered, or due to other factors such as the definition used for risk factors. We have used drinks per week to measure alcohol intake, while previous studies have used units of alcohol per week [28]. For drinking and smoking, misclassification bias may occur, which may be differential misclassification if this differs between participants with and without exposure to maltreatment. Integrating the five types of childhood maltreatment into a score, while relevant in the light of dose-response relationships between childhood maltreatment and CVD, may miss maltreatment type-specific effects. Nonetheless, previous work in this cohort has shown there is limited heterogeneity in the associations with CVD across maltreatment types [13].

Reports of sex differences in the association between childhood maltreatment subtypes and CVD have been mixed [6, 13, 41–44] however we did not identify any sex-specific differences. Despite UK Biobank being the largest available study with data on both childhood maltreatment and genetic data (required to derive polygenic scores), it is possible our analyses were still underpowered to detect evidence of effect modification. Replication and validation using other cohorts may elucidate whether we missed modification and sex-specific effects due to lack of statistical power.

Positive associations between childhood maltreatment and CVD subtypes and BMI have been reported extensively [5, 45] but associations with other CVD risk factors such as blood pressure and lipids are less established [45]. We replicated previously observed associations between maltreatment increasing BMI [41–43], smoking behaviour [6, 46], and decreasing SBP [47, 48].

## Interpretability of interactions and effect modifications

Interaction and effect modification are terms commonly used interchangeably despite representing subtly different concepts. The underlying statistical model is identical [49].

In an interaction, the effects of two causal and independent risk factors are studied, to assess whether their joint exposure leads to a different outcome compared with their independent associations. In effect modification, the effect of one exposure differs (or is modified) according to the value of a second exposure. These exposures are not necessarily causal [18]. Both interaction and effect modification can be estimated on the additive or the multiplicative scale. If two exposures are associated with the outcome, an interaction should be detected on at least one of the scales given sufficient statistical power.

In the case of childhood maltreatment, although consistent associations with CVD exist, causality is difficult to assess and therefore the term effect modification better reflects the model considered here. We investigated whether the effect of polygenic susceptibility to a certain trait differed according to values of the childhood maltreatment score. Previous studies

have typically only tested the additive scale for interaction [15, 16, 19, 39]. Because effect modification can occur on either the additive scale or the multiplicative scale, we analysed both scales [18, 19].

## Relevance for public health

The size of the interaction terms of the few effect modifications detected in this study were very small, even for models employing PGSs with SNPs included using a less stringent significance thresholds. This implies that the clinical impact of these effect modifications is very low, suggesting gene*environment effect modifications are not a major contributor to the excess burden of cardiovascular disease experienced by people who have experienced child maltreatment.

## Strengths and limitations

The major strength of our study is the use of a single large cohort, the UK Biobank, giving a large sample size and homogenous measures for our variables. While many studies rely on self-reported measures of phenotypes or disease outcomes, we used hospital and mortality records for CVD and calibrated physical measurements to define phenotypes. Reverse causality can bias analyses, whereby the diagnosis of CVD may lead to medical treatment or altered lifestyle behaviours altering the relative importance of the genotype on disease risk. We therefore excluded all participants with prevalent cases of CVD.

Another strength of this study lies in the use of PGSs rather than a single candidate gene approach, where candidate gene interactions have been shown to be spurious [50]. We also repeated analyses using PGSs generated at different significance thresholds, as PGSs including variants detected at less stringent P value thresholds may be more predictive of disease [51]. However, sample overlap was present in the AF GWAS used to derive the AF PGS, where UKB contributed to 60% to the sample, which may lead to overestimated effect sizes. Finally, PGSs are not subject to the same confounding as environmental exposures because they are determined at conception, though they may be affected by population structure. All models were adjusted for all 40 genetic PCs calculated by the UK Biobank to mitigate this bias.

A number of limitations remain. Participation in the mental health questionnaire, an inclusion criterion for our study, has been associated with lower BMI, higher socioeconomic position, and having no T2DM or depression, compared to the rest of the UK Biobank cohort [52]. Estimates may be biased by selection bias, where UK Biobank participants are more likely to be female, of higher socioeconomic position, less likely to be obese, to smoke, to drink alcohol on a frequent basis and to experience CVD compared to the rest of the population [24].

Childhood maltreatment was determined using self-reported retrospective measures, which could introduce recall bias. An alternative approach would be verifying self-reports with records from social services or other child protecting entities. However, only a fraction of maltreatment cases is reported [53] and thus many cases might be missed. As with most large studies, the measures used to construct the summary score were limited to type of maltreatment and self-reported frequency and thus do not capture aspects which may affect the severity of these exposures and their effect on health over the lifecourse, such as their timing in childhood, severity or duration. Another limitation is that despite the adjustment for maternal smoking and number of siblings, these might not be the most comprehensive indicators of childhood SEP. Maternal smoking may not be a robust indicator of SEP for this study's participants as the social patterning of maternal structure was different in the time these participants were born in comparison to current day. This means residual confounding, by childhood SEP or other unmeasured sources, may have biased our findings.

We applied a Bonferroni correction when considering the effect modification P values to address the risk of type 1 error due to multiple tests. This may be overly conservative as cardiovascular risk factors/diseases are correlated phenotypes. However, considering a less stringent P value threshold of P<0.05, there was still little evidence of an interaction between childhood maltreatment and any outcome other than BMI.

Finally, there are caveats with PGSs. Residual confounding by population structure may remain even after adjusting for 40 genetic PCs [54]. While our study only included participants of White British Ancestry and used summary statistics from similar populations, some of the PGSs explained a low proportion of the trait's variance. More so, our findings have limited generalizability to non-White European populations, as PGSs have been found to have very limited portability across ancestries [55]. The disproportionate scrutinising of a single ancestral background is an impediment in our understanding of CVD aetiology and the clinical application of PGSs. Indeed, 83% of GWASs released between 2005 and 2018 were on individuals of White European Ancestry, underscoring the importance of the need for trans-ethnic GWASs and the replication of studies such as ours in other populations.

## Conclusions

In this analysis on White British individuals from the UK Biobank, we exposure to childhood maltreatment can moderately exacerbate the effects of a genetic propensity to higher BMI. However, we found little evidence that childhood maltreatment modified the effects of other cardiovascular polygenic scores. Our results suggest gene*environment interactions are not a major contributor to the excess burden of CVD experienced by people who have experienced child maltreatment.

## Supporting information

**S1 Checklist. STROBE statement—checklist of items that should be included in reports of observational studies.**
(DOCX)

**S1 File.**
(DOCX)

## Acknowledgments

Quality Control filtering of the UK Biobank data was conducted by R.Mitchell, G.Hemani, T. Dudding, L.Corbin, S.Harrison, L.Paternoster as described in the published protocol (doi: 10. 5523/bris.1ovaau5sxunp2cv8rcy88688v). The MRC IEU UK Biobank GWAS pipeline was developed by B.Elsworth, R.Mitchell, C.Raistrick, L.Paternoster, G.Hemani, T.Gaunt (doi: 10. 5523/bris.pnoat8cxo0u52p6ynfaekeigi).

## Author Contributions

**Conceptualization:** Ana Gonçalves Soares, Abigail Fraser, Laura D. Howe, Alice R. Carter.

**Data curation:** Helena Urquijo, Ana Gonçalves Soares, Alice R. Carter.

**Formal analysis:** Helena Urquijo.

**Investigation:** Helena Urquijo.

**Supervision:** Ana Gonçalves Soares, Abigail Fraser, Laura D. Howe, Alice R. Carter.

**Writing – original draft:** Helena Urquijo.

**Writing – review & editing:** Helena Urquijo, Ana Gonçalves Soares, Abigail Fraser, Laura D. Howe, Alice R. Carter.

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
