## [Decision Letter · Decision Letter 0]

19 Dec 2022

PONE-D-22-30912Investigating effect modification between childhood maltreatment and genetic risk for cardiovascular disease in the UK BiobankPLOS ONE

Dear Dr. Urquijo,

Thank you for submitting your manuscript to PLOS ONE. After careful consideration, we feel that it has merit but does not fully meet PLOS ONE’s publication criteria as it currently stands. Therefore, we invite you to submit a revised version of the manuscript that addresses the points raised during the review process.

We look forward to receiving your revised manuscript.

Kind regards,

Heming Wang, PhD

Academic Editor

PLOS ONE

Journal Requirements:

Reviewers' comments:

Reviewer's Responses to Questions

**Comments to the Author**

1. Is the manuscript technically sound, and do the data support the conclusions?

Reviewer #1: Yes

Reviewer #2: Yes

2. Has the statistical analysis been performed appropriately and rigorously? 

Reviewer #1: I Don't Know

Reviewer #2: Yes

3. Have the authors made all data underlying the findings in their manuscript fully available?

Reviewer #1: Yes

Reviewer #2: Yes

4. Is the manuscript presented in an intelligible fashion and written in standard English?

Reviewer #1: Yes

Reviewer #2: Yes

5. Review Comments to the Author

Reviewer #1: This is a great paper though there was no finding on significant interaction between childhood maltreatment and PGS influencing CVD.

1. There may be no interaction indeed, however, a true association may have not been revealed due to low statistical power using the 1 df test of the interaction effect alone. The 2 df test is more powerful when PGS is weak, and the interaction effect is moderate than either the 1df test of the PGS or the 1df test of the interaction alone. Using the 2df joint test under the H0: Beta(PGS) = Beta(PGX x childhood maltreatment), you may be able to identify associations that were not discovered using the 1df test.

2. Have you considered including age squared as one of the covariates? In case including age only in the model would not be sufficient to control for the effect of age.

3. A minor suggestion is that I found some typos or grammatical errors in the context. For example, in the method for effect modification, I was wondering whether you meant multivariable “logistic” regression for binary traits, not “linear” regression.

Reviewer #2: - Statistical significance of the interaction (p-value) should be mentioned in the abstract.

- Table 2 presented the variable distributions in the analytical sample versus the whole UKB to show that the distributions of various factors were not substantially different. This information is better placed into supplementary materials, since the information of the whole UKB was not directly related to this study and is very likely available from other studies that used the full cohort. Instead, showing the sample characteristics by the childhood maltreatment score or by sex would be more informative.

- While it makes sense to analyze the continuous traits as per SD difference, the results would be more interpretable if the SD can be converted to the actual units (e.g., how much is per SD of BMI in this analytical sample?)

- Childhood maltreatment score was used as a continuous variable in all analyses. The underlying hypothesis is that different types of maltreatment have the same equal effect on cardiovascular traits, which may not be true. The authors did not examine potential heterogeneity in the associations by maltreatment types although the data are available (e.g., sexual abuse is often considered the more severe form of childhood maltreatment).

- The study only found a significant interaction between childhood maltreatment and genetic susceptibility to BMI. A recent study reported that BMI was the most important factor that mediated the association between childhood abuse and metabolic profiles in midlife (PMID: 35471987). Incorporating this study in the discussion may potentially increase the significance of the findings from the current study.

- Discussion: it should be acknowledged that the assessment of childhood maltreatment was also crude that did not capture frequency, duration, severity or timing for each type of maltreatment. For example, childhood maltreatment that occurs earlier in childhood may result in stronger gene-environment interaction effect. Also, the definition of ‘child’ in the assessment question is vague. Someone could consider an “adolescent” as a child while others may not. The maltreatment during adolescence may not have the same biological effect as the one during early childhood <5 years old.

6. PLOS authors have the option to publish the peer review history of their article (what does this mean?). If published, this will include your full peer review and any attached files.

Reviewer #1: No

Reviewer #2: No

---

## [Author Response · Author response to Decision Letter 0]

23 Feb 2023

All authors would like to express sincere appreciation for the careful evaluation of our research article, “Investigating effect modification between childhood maltreatment and genetic risk for cardiovascular disease in the UK Biobank”, for publication in Plos One. After having considered the comments in detail, which found them to be highly constructive and helpful, we have revised the manuscript accordingly. We put our best effort into incorporating every suggestion. We find the revisions to have improved the work considerably and strengthened its conclusions. Full responses to all feedback together with corresponding changes to the manuscript are provided in the Response to Reviewers letter.

---

## [Decision Letter · Decision Letter 1]

19 Apr 2023

Investigating effect modification between childhood maltreatment and genetic risk for cardiovascular disease in the UK Biobank

PONE-D-22-30912R1

Dear Dr. Urquijo,

We’re pleased to inform you that your manuscript has been judged scientifically suitable for publication and will be formally accepted for publication once it meets all outstanding technical requirements.

Kind regards,

Heming Wang, PhD

Academic Editor

PLOS ONE

Additional Editor Comments (optional):

Reviewers' comments:

Reviewer's Responses to Questions

**Comments to the Author**

1. If the authors have adequately addressed your comments raised in a previous round of review and you feel that this manuscript is now acceptable for publication, you may indicate that here to bypass the “Comments to the Author” section, enter your conflict of interest statement in the “Confidential to Editor” section, and submit your "Accept" recommendation.

Reviewer #1: All comments have been addressed

2. Is the manuscript technically sound, and do the data support the conclusions?

Reviewer #1: Yes

3. Has the statistical analysis been performed appropriately and rigorously? 

Reviewer #1: Yes

4. Have the authors made all data underlying the findings in their manuscript fully available?

Reviewer #1: Yes

5. Is the manuscript presented in an intelligible fashion and written in standard English?

Reviewer #1: Yes

6. Review Comments to the Author

Reviewer #1: Thank you for clarifying and providing the details with sufficient evidence. I agree with all author's comments.

7. PLOS authors have the option to publish the peer review history of their article (what does this mean?). If published, this will include your full peer review and any attached files.

Reviewer #1: No

---

## [Editor Report · Acceptance letter]

25 Apr 2023

PONE-D-22-30912R1 

Investigating effect modification between childhood maltreatment and genetic risk for cardiovascular disease in the UK Biobank 

Dear Dr. Urquijo:

I'm pleased to inform you that your manuscript has been deemed suitable for publication in PLOS ONE. Congratulations! Your manuscript is now with our production department. 

Kind regards, 

on behalf of

Dr. Heming Wang 

Academic Editor

PLOS ONE